# Bloodstream Infections in Intensive Care Unit during Four Consecutive SARS-CoV-2 Pandemic Waves

**DOI:** 10.3390/antibiotics12091448

**Published:** 2023-09-14

**Authors:** Giacomo Pozza, Giacomo Casalini, Cosmin Lucian Ciubotariu, Andrea Giacomelli, Miriam Galimberti, Martina Zacheo, Andrea Rabbione, Margherita Pieruzzi, Letizia Oreni, Laura Galimberti, Riccardo Colombo, Giuliano Rizzardini, Cristina Pagani, Sara Giordana Rimoldi, Cecilia Bonazzetti, Anna Lisa Ridolfo, Spinello Antinori

**Affiliations:** 1III Division of Infectious Diseases, Luigi Sacco Hospital, ASST Fatebenefratelli Sacco, 20157 Milan, Italy; giacomo.pozza@unimi.it (G.P.); giacomo.casalini@unimi.it (G.C.); cosmin.ciubotariu@unimi.it (C.L.C.); miriam.galimberti@unimi.it (M.G.); martina.zacheo@unimi.it (M.Z.); andrea.rabbione@unimi.it (A.R.); margherita.pieruzzi@unimi.it (M.P.); letizia.oreni@alice.it (L.O.); laura.galimberti@asst-fbf-sacco.it (L.G.); annalisa.ridolfo@asst-fbf-sacco.it (A.L.R.); spinello.antinori@unimi.it (S.A.); 2Department of Biomedical Sciences and Clinics, Università degli Studi di Milano, 20157 Milan, Italy; 3Intensive Care Unit, ASST Fatebenefratelli Sacco, 20157 Milan, Italy; riccardo.colombo@asst-fbf-sacco.it; 4I Division of Infectious Diseases, Luigi Sacco Hospital, ASST Fatebenefratelli Sacco, 20157 Milan, Italy; giuliano.rizzardini@asst-fbf-sacco.it; 5Clinical Microbiology, Virology and Bioemergency, Luigi Sacco Hospital, ASST Fatebenefratelli Sacco, 20157 Milan, Italy; cristina.pagani@asst-fbf-sacco.it (C.P.); sara.rimoldi@asst-fbf-sacco.it (S.G.R.); 6Infectious Diseases Unit IRCCS, Policlinico Sant’Orsola, Department Medical Surgical Science, University of Bologna, 40138 Bologna, Italy; cecilia.bonazzetti@unibo.it

**Keywords:** COVID-19, bloodstream infection, critically ill, enterococcus, pseudomonas

## Abstract

Critically ill COVID-19 patients are at an increased risk of bloodstream infections (BSIs). We performed a retrospective observational single-center study on COVID-19 patients admitted to intensive care unit (ICU) to assess the incidence of BSIs in four consecutive periods: 21 February–31 July 2020 (W1), 1 August 2020–31 January 2021 (W2), 1 February–30 September 2021 (W3) and 1 October 2021 and 30 April 2022 (W4). BSIs that occurred 48 h after ICU admission were included. The crude incidence of BSIs was estimated by means of Poisson distribution normalized to 1000 patient-days. A total of 404 critically ill COVID-19 patients were admitted to ICU, of whom 284 (61%) developed at least one episode of BSI with an overall crude incidence of 87 events every 1000 patient-days (95% CI 77–98) without a significant difference in consecutive epidemic periods (*p* = 0.357). Gram-positive bacteria were the most frequent etiological agents of BSIs, contributing to 74.6% episodes. A progressive decrease in BSIs due to *Enterococcus* spp. was observed (W1 57.4%, W2 43.7%, W3 35.7% and W4 32.7%; *p* = 0.004). The incidence of BSIs remained stable during different epidemic periods. *Enterococcus* spp. prevalence was significantly reduced, although still accounted for one third of BSIs in more recent epidemic periods.

## 1. Introduction

The COVID-19 pandemic determined an abrupt increase in critically ill patients requiring intensive care unit (ICU) assistance [1,2]. This was particularly true for Italy, and in particular for the Lombardy region where, starting from 20 February 2020, a dramatic increase in patients with SARS-CoV-2 infection with severe pneumonia requiring respiratory support was observed in a few weeks [1,2]. Such a rapid increase in the demand for respiratory support and ICU admittance determined a rapid reorganization of the health care system and human resource reallocation, posing a serious challenge to the infection control measures due to the pandemic contingency. Moreover, the natural history of COVID-19 predisposes critically ill COVID-19 patients to a prolonged ICU stay, exposing them to the risk of developing secondary infections including bloodstream infections (BSIs) and ventilator-associated lower respiratory tract infections (VA-LRTI). Several studies have found an increased incidence of secondary BSIs in critically ill COVID-19 patients and, in some studies, these infections were associated with more severe initial presentation, prolonged hospital course, and worse clinical outcomes [3,4,5,6,7,8]. In particular, in a multicenter study conducted by our group, we found that the Sequential Organ Failure Assessment (SOFA) score and Charson Comorbidity index were both associated with an increased risk of BSI development in critically ill COVID-19 patients [2]. Moreover, as mentioned above, practices of antimicrobial stewardship were substantially reduced during the first pandemic period, especially in the very early days of the pandemic when health care systems were overwhelmed by an unprecedented surge of critically ill patients [9,10], together with an increase in the use of broad-spectrum antibiotics [11]. Preliminary reports during the first pandemic periods suggested a possible increase in BSIs caused by *Enterococcus* spp. [12,13,14], which was not fully explained by nosocomial transmission when whole-genome sequencing was applied [15]. Soon after the first pandemic period, the characteristics of the pandemic evolved due to several reasons. First, the SARS-CoV-2 viral evolution determined the selection of SARS-CoV-2 variants of concern (VOCs) with an increased transmissibility but that are also, after the advent of the Omicron variant and subsequent sub-variants, characterized by less pathogenicity [16,17]. Second, improvement in patient care was significant with an increased knowledge of SARS-CoV-2 infection’s natural history and the availability of drugs for different stages of COVID-19 [18]. Third, the pivotal change during the pandemic was driven by the mass administration of the COVID-19 primary vaccine cycle which was able to reduce the risk of progression to severe COVID-19 with a high vaccine effectiveness [19]. Most of what is known regarding the incidence and epidemiology of BSIs in critically ill COVID-19 patients derives from studies conducted during the first two epidemic periods [20,21,22,23], whereas little is known regarding the last pandemic periods in terms of the incidence and epidemiology of BSIs.

We performed a retrospective observational study to assess the incidence of BSIs during four consecutive pandemic periods and to characterize the epidemiology of BSIs.

## 2. Results

### 2.1. Characteristics of the Study Population

Four hundred and four patients critically ill COVID-19 patients were admitted to ICU for at least 48 h during the study period. Of these, 109 (27%) were females and the median age was 63 years (IQR 56–70). The characteristics of the study population, according to being admitted in different epidemic periods, are reported in Table 1. A total of 344 (85.1%) patients required invasive mechanical ventilation and 174 (43.1%) died. Patients admitted during W2, W3, and W4 were older compared to subjects admitted during W1 [median years 65 (IQR 58–72), 64 (IQR 56–72), 64 (IQR 58–69) vs. 61 (IQR 50–69), *p* = 0.018, respectively]. The proportion of subjects with obesity was higher during W3 and W4 compared to W1 and W1 (44.1% and 42.4% vs. 26.3% and 34.3%, *p* = 0.040, respectively). No significant differences in other comorbidities (diabetes, pneumological comorbidities, cardiovascular comorbidities, nephrological comorbidities, oncological comorbidities, and liver disease) were observed across different epidemic periods. The median SOFA score at ICU admission was higher during W1 and W2 compared to W3 and W4 (9 (IQR 7–12) and 9 (IQR 7–10) vs. 7 (IQR 3–8) and 5 (IQR 3–7), *p* < 0.001, respectively). COVID-19 therapies evolved overtime with an increase in steroids (*p* < 0.001) and heparin use (*p* < 0.001), and a reduction in tocilizumab (*p* < 0.001) and remdesivir (*p* < 0.001) use. As expected, no subject received a COVID-19 vaccination before ICU admission during W1 and W2. On the contrary, the proportion of subjects without a previous COVID-19 vaccination reduced from W3 to W4 (87% vs. 54.5%) and 12.1% of patients hospitalized in an ICU during W4 received a primary vaccination cycle plus a booster dose before hospital admission. No significant difference was observed in median time from symptoms onset to ICU admission across different waves (*p* = 0136). A trend toward a faster access to ICU once hospitalized for severe COVID-19 was observed during W3 and W4 compared to W1 and W2 (median days 1 (IQR 0–3) and 1 (IQR 0–5) vs. 2 (IQR 0–5) and 2 (IQR 0–5), *p* = 0.067, respectively). No significant differences were observed across different waves in terms of the length of ICU stay (*p* = 0.976) and the proportion of ICU-admitted mechanically ventilated subjects (*p* = 0.423), whereas a significant reduction in in-hospital mortality was observed from W1 to W4 (53.5% vs. 31.8%; *p* < 0.001).

### 2.2. Bloodstream Infections Incidence

A total of 284 (61%) subjects developed at least one episode of BSI during their ICU stay, with 489 total episodes recorded. Overall, the crude incidence was 87 events every 1000 patient-days at risk (95% CI 77–98), without a significant difference in consecutive epidemic periods (*p* = 0.357). The overall cumulative incidence of BSIs at 15 days from ICU admission was 58.2% (95% CI 54–63) and no statistically significant difference was observed between different waves [W1 59.6% (95% CI 50.7–70), W2 63.2% (95% CI 55.6–71.9), W3 50.5% (95% CI 41.8–61) and W4 57.7% (95% CI 47–70.9); *p* = 0.404] (Figure 1).

### 2.3. Epidemiology of BSI

The microbiological characterization of BSI episodes according to different epidemic periods is reported in Table 2. Gram-positive bacteria were the most frequent etiological agents of BSIs across all pandemic waves, contributing to 74.6% of BSI episodes. A progressive decrease in BSIs due to *Enterococcus* spp. was observed from W1 (57.4%) to W4h (32.7%) (*p* = 0.004), while the frequency of coagulase-negative staphylococci (CoNS) increased from W1 (26.6%) and W2 (20.4%) to W3 (38.6%) and W4 (33.9%) (*p* = 0.003). Regarding multi-drug-resistant gram-positive bacteria, vancomycin-resistant enterococci (VRE) accounted for 4.9% of BSI episodes and methicillin-resistant *Staphylococcus aureus* (MRSA) for 3.1% of BSI episodes. Gram negative bacteria contributed to 40% of BSI episodes. The most common pathogens were *Enterobacterales* (21.8%) with 4.7% of episodes being caused by extended spectrum beta lactamases (ESBL) bacteria and 2.9% of episodes being caused by carbapenemase-producing *Enterobacterales* (CPE). Regarding CPE, a progressive reduction was observed from W1 (10.6%) to W4 (0%) (*p* < 0.001). *Pseudomoas aeruginosa* accounted for 10.4% of BSI episodes, with 2.9% of episodes being caused by MDR strains. BSIs caused by *P aeruginosa* significantly increased across the last three pandemic waves (W1 3.2%, W2 9%, W3 15.2% and W4 18%; *p* = 0.007) without a concomitant surge in the number of MDR strains. Yeasts accounted for 2% of BSI episodes, with *Candida albicans* (seven episodes) as the most common causative agent followed by *C. tropicalis* (two episodes), *C. glabrata* (one episode) and *C. parapsilosis* (one episode).

## 3. Discussion

In our study, 61% of critically ill COVID-19 patients experienced at least one BSI episode. Most of the BSI episodes were caused by gram positive bacteria (74.6%) with gram negative contributing to 40%, and yeasts to 2% of BSI episodes. Although BSI incidence remained stable during different epidemic periods, a progressive decrease in *Enterococcus* spp. episodes and an increase in episodes caused by CoNS and *P. aeruginosa* were observed.

Studies examining secondary infections in critically ill COVID-19 patients are heterogeneous but they show a substantially increased incidence of BSIs, and this is also confirmed by a systematic meta-analysis with an estimated occurrence of 29.6% (95% CI 21.7–38.8%) [24]. The higher prevalence of BSIs observed in our study could have several potential explanations. First, Bonazzetti et al. previously reported a higher sampling of blood cultures at our center when compared to others in critically ill COVID-19 patients admitted to ICUs [2]. In particular, in the same study, the authors found a significant discrepancy between centers in terms of blood culture collections normalized per days of ICU stay, suggesting a different threshold for blood culture request in different Italian centers which could, in part, explain the high BSI cumulative incidence observed at our center. Second, a reduction in infection control practices determined by the overwhelming pressure on the health care system could have occurred during the pandemic period, challenging the virtuous norms of infection control. In particular, the first two pandemic waves were the most critical at our center, with a shortage of sterile gowns; some even had to be reused during the first pandemic wave [12]. In addition, if, on the one hand, all the available single rooms at our center had a dedicated medical cart containing all of the equipment required for medical and nursing procedures, on the other, the two-bedded rooms were equipped with only one medical cart for both patients. In addition, the human resources to perform nosocomial infection control and antibiotic stewardship were significantly reduced during the study period and the routine surveillance for multi-resistant agents (nasal, cutaneous and rectal swabs upon admission and during hospitalization), and the consequent eventual subsequent isolation of those who tested positive, had to be suspended during the most critical days of the pandemic [12]. Moreover, during the first SAR-CoV-1 outbreak in 2003, it was shown that, despite the mandatory use of respiratory and contact precautions, a paradoxical increase in the incidence rate of MRSA infections occurred [25]. Third, the intrinsic characteristics of SARS-CoV-2 infection determines several factors which could predispose individuals with COVID-19 to an increased risk of nosocomial infections and, in particular, of BSIs. In particular, the prolonged time of ICU stay, the severity of lung involvement, the associated state of immune-dysregulation induced by the disease and by the medication used to counteract the respiratory disfunction and the burden of comorbidities are all predisposing factors to an increased incidence of BSIs.

The incidence, type, and etiology of co-infections and superinfections in patients hospitalized for COVID-19 during the first epidemic periods have been summarized in two large systematic reviews, showing a prevalence of superinfections up to 41% in critically ill COVID-19 patients requiring ICU assistance [24,26]. This was quite a different scenario when compared to the latest pre-pandemic epidemiological report by the European Centre for Disease Control and Prevention (ECDC) on healthcare-associated infections in ICU (2017), in which a BSI rate of 3.7% (crude incidence of 1.9/1000 patient-days at risk) was reported [27]. The reported pre-pandemic incidence of BSIs in ICUs was extremely low when compared to that observed during the first two pandemic periods by the COVID-ICU study group (10.3 BSI every 1000 patient day at risk) [28], by Grasselli G. et al. [29] (16.4 cases every 1000 patient days at risk), by Kurt and colleagues (50.2 cases every 1000 patient at risk) [30] and by Cataldo M.A. et al. (37.3 BSI every 1000 patient days at risk) [3]. The disproportionately high incidence of BSIs in the first pandemic periods could be explained not only by the unprecedented overwhelming pressure on health care systems but also by the characteristics of critical COVID-19: the disease severity at ICU admission (i.e., elevated SOFA score) [2], the severe immune deregulation [31] and the prolonged course of ARDS with a protracted requirement for mechanical ventilation [32]. These are characteristics of the natural history of the disease which predisposed COVID-19 subjects to an increased incidence of BSIs when compared to other viral pneumonia, i.e., influenza, in which the incidence of BSIs is far less [33].

In our study, we did not observe a reduction in the incidence of BSIs, although the disease severity at ICU admission significantly reduced during subsequent epidemic periods and, in parallel, a significant reduction in mortality was observed. There are some potential explanations for what we observed. First, although the disease severity at ICU admission was lower in more recent epidemic periods, subjects admitted were more prevalently obese [34]. In addition, the median time of ICU stay (11 days IQR 6–22) and median mechanical ventilation duration (11 days IQR 6–21) were substantially comparable between different epidemic periods, which are known to be among the strongest factors associated with secondary infection development in ICUs [35]. In addition, it is worth mentioning that the proportion of CoNS-related BSIs significantly increased in the last two epidemic periods (W3 and W4), suggesting that most of these episodes were catheter-related and with little impact on the overall disease course and clinical outcome [36].

Regarding the epidemiology of BSIs, gram-positive bacteria, and in particular *Enterococcus* spp., were predominant across all epidemic periods and, although a significant reduction in BSIs caused by *Enterococcus* spp. was observed, in more recent epidemic periods, *Enterococcus* spp. still accounted for one out of three BSI episodes. A high frequency of enterococcal BSIs in the first pandemic months was reported not only by our group [12], but also by others [13,14,15,30]. The high proportion of BSIs caused by *Enterococcus* spp. observed in more recent periods suggests that what has been described previously was not a chance finding. There are some potential explanations for what has been observed. First, a global rise in *Enterococcus* spp. as nosocomial pathogens had been already described before the COVID-19 pandemic [37]. This posed new challenges in the management of critically ill patients due to the potential occurrence of multi-drug-resistant strains of enterococci. In fact, at our center, VRE accounted for 4.9% of all BSI episodes. Second, a potential pathophysiological explanation for this observation relies on the disruption of the intestinal mucosal barrier due to SARS-CoV-2 infections, together with hemodynamic disorders in the mesenteric vascular district in critically ill COVID-19 patients [38]. It has been shown that SARS-CoV-2 is able to productively infect human enterocytes [39]. Thus, it is possible that SARS-CoV-2 infection increases intestinal permeability and predisposes to bacterial translocation, which is a known risk factor for the development of BSIs and, in particular, for bacteria which colonize the enteric tract. This has also been previously shown for SARS-CoV-1, with pathologic studies showing the enteric barrier damage induced by the virus [40] which was confirmed in the pathological findings of COVID-19 patients [41,42], although the direct SARS-CoV-2 infection of the enterocytes seems not to be the primary cause of severe gut damage in most COVID-19 patients [43].

The rate of *P. aeruginosa* BSI gradually increased during the study period from 3.2% of gram negative BSI in the first wave to 18% in the fourth wave, as reported by Amarsy et al. and Sloot et al. [38,44]. This may be partially explained by the occurrence of risk factors typically associated with *P. aeruginosa* infections, such as older age, multiple comorbidities and the massive lung involvement of COVID-19 patients. In addition, Rhoades et al. showed that the nasal microbiome of SARS-CoV-2-positive patients, at the time of diagnosis, when compared to uninfected controls, is characterized by an abundance of bacterial pathogens, including *P. aeruginosa*, which is also positively associated with SARS-CoV-2 RNA load. This abundance of bacterial pathogens in the nasal cavity could potentially contribute to an increased incidence of secondary bacterial infections, especially in those under mechanical ventilation where the primary source of *P. aeruginosa* BSIs could be identified in the lung superinfection [45].

Yeasts accounted for 2% of BSI episodes in our study. In particular, critically ill COVID-19 patients are at an increased risk of candidemia due to several reasons: (i) the wide use of broad spectrum antibiotics [11], (ii) the requirements of ICU admittance with in-dwelling vascular catheters [46] and (iii) the requirement, in the most severe cases, for iatrogenic immunosuppression [47]. In addition, the development of candidemia is associated with a very poor prognosis with a recent metanalysis estimating a pooled in-ICU mortality of 66.77% (95% CI, 57.70% to 74.75%) [48].

Our study has several limitations. First, it is a retrospective study and thus predisposed to missing data. Second, as mentioned above, the incidence of BSIs reported in our study represents what has been observed in one single reference center in northern Italy that was first affected by the pandemic contingencies and thus is not fully representative of other Italian clinical centers. Third, the high incidence of BSIs could be potentially related to the low threshold for requesting blood cultures by physicians in charge of the patients. Thus, the comparison of BSI incidence with other settings is not easy. Nevertheless, our study also has the strength of covering a long period characterized by significant improvement in COVID-19 management and the ability to catch the epidemiological modification occurred at our center.

## 4. Materials and Methods

### 4.1. Study Design, Setting and Participants

This was a retrospective, single-center, observational cohort study, which enrolled consecutive adult patients with critical COVID-19 hospitalized for at least 48 h at the ICU of the Luigi Sacco Hospital (ASST Fatebenefratelli Sacco, Milan, Italy) between the 21 February 2020, and the 7 April 2022. For the purpose of the present study, four consecutive SARS-CoV-2 epidemic periods, corresponding to pandemic waves in Italy, were chosen [49,50]: 21 February–31 July 2020 (first wave, W1), 1 August 2020–31 January 2021 (second wave, W2), and 1 February–30 September 2021 (third wave, W3) and, finally, 1 October 2021 and 30 April 2022 (forth wave, W4).

### 4.2. Data Collection, Procedures and Definitions

For each patient in the study, we collected demographic data (age, biological sex, nationality) and clinical data (type and number of comorbidities, COVID-19 vaccination status, SOFA score at ICU admission, ICU length of stay, need for orotracheal intubation, length of mechanical ventilation, concomitant therapies (heparin, steroids, remdesivir, tocilizumab and baricitinib), data of death and/or discharge. BSIs were defined using the Center for Disease and Control criteria [51]. Only BSIs and VA-LRTIs occurring >48 h from ICU admission were included in the study. The isolation of a common skin organism usually associated with contamination had to be corroborated by two sets of blood cultures to be considered a BSI [52]. ICU-acquired bacteremia was defined as a BSI if it was diagnosed  >48 h after ICU admission. To be considered a new episode a BSI, it had to meet the criteria for an ICU-acquired BSI due to a different organism 48 h after the initial infection. The EUCAST Expert Rules were used to define an isolated pathogen multi-drug r-sistant (MDR) [53].

### 4.3. Laboratory Procedures

The microbial species causing BSIs were identified using Vitek MS matrix-assisted laser desorption/ionization time-of-flight mass spectrometry (bioMérieux, Marcy l’Etoile, France). Antimicrobial susceptibility and resistance detection of the clinical isolates were determined using the automated Vitek 2 system (bioMérieux, Bagno a Ripoli, Florence, Italy). The interpretation of susceptibility patterns was performed according to the European Committee on Antimicrobial Susceptibility Testing [53].

### 4.4. Outcomes

The primary outcome was microbiologically proven BSI.

### 4.5. Statistical Analysis

Continuous variables are reported as means with standard deviations or medians and interquartile ranges (IQR). Categorical variables are expressed as percentages. Continuous variables were compared using the Student’s *t*-test or the Mann–Whitney U test, as appropriate. Categorical variables were compared using the chi-square test or Fisher’s exact test. The crude incidence of BSIs was estimated as the number of incident first occurrences of the infectious episodes divided by the patient-days at risk (days in the ICU), with a corresponding 95% confidence interval (CI), computed using a Poisson distribution. Incidence rates were normalized to 1000 patient-days. Cumulative incidence was calculated using death and discharge as competing events. A Gray test was used to assess differences in cumulative incidence between different epidemic periods. All the statistical tests were two-tailed and were considered significant with a *p* less than 0.05.

## 5. Conclusions

We found a high rate of BSIs in critically ill COVID-19 patients, without a substantial decline in more recent epidemic periods. *Enterococcus* spp., although significantly reduced, also remained predominant in more recent periods, confirming the prelaminar observations during the first pandemic wave and suggesting the underlying pathophysiological mechanism which predisposes critically ill COVID-19 subjects to enterococcal infections.

## Figures and Tables

**Figure 1 antibiotics-12-01448-f001:**
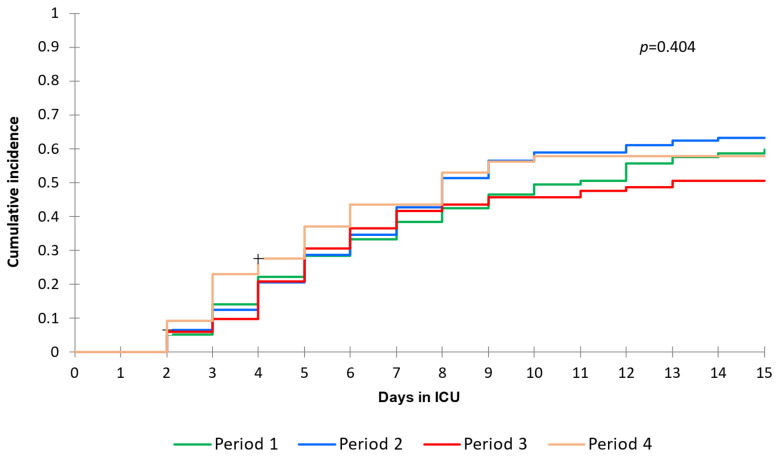
Cumulative incidence of BSIs according to being admitted to ICU in different SARS-CoV-2 epidemic periods (21 February–31 July 2020 (period 1), 1 August 2020–31 January 2021 (period 2), 1 February–30 September 2021 (period 3) and 1 October 2021 and 30 April 2022 (period 4).

**Table 1 antibiotics-12-01448-t001:** Demographical and clinical characteristics of the study population.

Characteristics	Overall n = 404	W1 n = 99	W2 n = 137	W3 n = 102	W4 n = 66	*p*-Value
**Age, median (IQR)**	63 (56, 70)	61 (50, 69)	65 (58, 72)	64 (56, 72)	64 (58, 69)	0.018
**Female sex at birth, n (%)**	109 (27.0)	22 (22.2)	33 (24.1)	31 (30.4)	23 (34.8)	0.222
**Italian, n (%)**	317 (78.5)	83 (83.8)	103 (75.2)	73 (71.6)	58 (87.9)	0.031
**Comorbidities, n (%)**						
Obesity	146 (36.1)	26 (26.3)	47 (34.3)	45 (44.1)	28 (42.4)	0.040
Pneumological comorbidities	38 (9.4)	5 (5.1)	19 (13.9)	7 (6.9)	7 (10.6)	0.098
Cardiovascular comorbidities	218 (54.0)	49 (49.5)	83 (60.6)	49 (48.0)	37 (56.1)	0.189
Metabolic diseases	151 (37.4)	31 (31.3)	55 (40.1)	35 (34.3)	30 (45.5)	0.235
Diabetes	73 (18.1)	13 (13.1)	25 (18.2)	18 (17.6)	17 (25.8)	0.233
Nephrological comorbidities	28 (6.9)	8 (8.1)	10 (7.3)	4 (3.9)	6 (9.1)	0.544
Oncological comorbidities	35 (8.7)	6 (6.1)	11 (8.0)	11 (10.8)	7 (10.6)	0.612
Immunological comorbidities	27 (6.7)	10 (10.1)	7 (5.1)	4 (3.9)	6 (9.1)	0.235
Liver diseases	5 (1.2)	0 (0.0)	3 (2.2)	2 (2.0)	0 (0.0)	0.318
**Number of comorbidities, n (%)**						
0	117 (29.0)	30 (30.3)	38 (27.7)	33 (32.4)	16 (24.2)	0.228
1	132 (32.7)	37 (37.4)	35 (25.5)	37 (36.3)	23 (34.8)	
2	107 (26.5)	25 (25.3)	43 (31.4)	23 (22.5)	16 (24.2)	
3+	48 (11.9)	7 (7.1)	21 (15.3)	9 (8.8)	11 (16.7)	
**SOFA score, median (IQR)**	8 (4, 10)	9 (7, 12)	9 (7, 10)	7 (3, 8)	5 (3, 7)	<0.001
**Treatments received, n (%)**						
Tocilizumab	45 (11.1)	38 (38.4)	1 (0.7)	6 (5.9)	0 (0.0)	<0.001
Remdesivir	109 (27.0)	49 (49.5)	21 (15.3)	23 (22.5)	16 (24.2)	<0.001
Heparin	311 (77.0)	42 (42.4)	122 (89.1)	91 (89.2)	56 (84.8)	<0.001
Steroids	277 (68.6)	19 (19.2)	130 (94.9)	75 (73.5)	53 (80.3)	<0.001
**N° vaccine doses anti-SARS-CoV-2, n (%)** **W3 (n = 100) W4 (n = 66)**						
0	-	-	-	87 (87.0)	36 (54.5)	
1	-	-	-	10 (10.0)	7 (10.6)	
2	-	-	-	3 (3.0)	15 (22.7)	
3	-	-	-	0 (0.0)	8 (12.1)	
**Days from symptoms to ICU access, median (IQR)**	11 (7, 14)	11 (8, 15)	10 (7, 13)	11 (7, 14)	10 (8, 13)	0.136
**Days from hospitalization to ICU access, median (IQR)**	1 (0, 5)	2 (0, 5)	2 (0, 5)	1 (0, 3)	1 (0, 5)	0.067
**Length of stay in ICU, median (IQR)**	11 (6, 22)	12 (7, 17)	11 (5, 22)	11 (5, 22)	11 (6, 24)	0.976
**MV requirement, n** **(%)**	344 (85.1)	87 (87.9)	118 (86.1)	87 (85.3)	52 (78.8)	0.423
**Length of MV (n =** **344), median (IQR)**	11 (6, 21)	11 (7, 18)	11 (6, 24)	11 (5, 20)	10 (6, 20)	0.881
**Mortality, n (%)**	174 (43.1)	53 (53.5)	70 (51.1)	30 (29.4)	21 (31.8)	<0.001

List of abbreviations: n, number; IQR, inter quartile range; ICU, intensive care unit; SOFA, Sequential Organ Failure Assessment; MV, mechanical ventilation; W1, 21 February–31 July 2020; W2, 1 August 2020–31 January 2021; W3, 1 February–30 September 2021, W4, 1 October 2021 and 30 April 2022.

**Table 2 antibiotics-12-01448-t002:** Types of isolates from BSI episodes.

	Overall n (%)	W1 n (%)	W2 n (%)	W3 n (%)	W4 n (%)
	489 (100)	94 (100)	222 (100)	112 (100)	61 (100)
**Gram positive**	365 (74.6)	74 (78.7)	159 (71.6)	88 (78.6)	44 (72.1)
*Enterocuccus* spp.	211 (43.1)	54 (57.4)	97 (43.7)	40 (35.7)	20 (32.7)
VRE	24 (4.9)	5 (5.3)	14 (6.3)	5 (4.5)	0 (0)
*Staphylococcus aureus*	40 (8.2)	7 (7.4)	17 (7.6)	9 (8)	7 (11.5)
MRSA	15 (3.1)	5 (5.3)	6 (2.7)	2 (1.8)	2 (3.3)
CoNS	136 (27.8)	25 (26.6)	46 (20.7)	44 (39.2)	21 (34.4)
*Viridans* group Streptococci	10 (2)	0 (0)	4 (1.8)	3 (2.7)	3 (4.9)
**Gram negative**	196 (40)	29 (30.8)	101 (45.5)	41 (36.6)	25 (40.9)
Enterobacterales	108 (22)	19 (20.2)	52 (23.4)	21 (18.7)	16 (26.2)
ESBL+	23 (4.7)	6 (6.3)	4 (1.8)	7 (6.2)	6 (9.8)
CPE	14 (2.9)	10 (10.6)	3 (1.3)	1 (1)	0 (0)
*Enterobacter* spp.	47 (9.6)	6 (6.4)	31 (13.9)	5 (4.5)	5 (8.2)
*Pseudomonas aeruginosa*	51 (10.4)	3 (3.2)	20 (9)	17 (15.2)	11 (18)
MDR	14 (2.9)	1 (1.1)	7 (3.1)	5 (4.5)	1 (1.6)
*Strenotrophomonas maltophilia*	2 (0.4)	1 (1.1)	1 (0.4)	0 (0)	0 (0)
*Acinetobacter* spp.	6 (1.2)	1 (1.1)	2 (1)	2 (1.8)	1 (1.6)
**Yeasts**	10 (2)	3 (3.2)	3 (1.3)	1 (0.9)	3 (4.9)
*C. albicans*	7 (1.4)	3 (3.2)	3 (1.3)	1 (0.9)	0 (0)
*C. glabrata*	1 (0.2)	0 (0)	0 (0)	0 (0)	1 (1.6)
*C. tropicalis*	2 (0.4)	0 (0)	0 (0)	1 (0.9)	1 (1.6)
*C. parapsilosis*	1 (0.2)	0 (0)	0 (0)	0 (0)	1 (1.6)

List of abbreviations: VRE, vancomycin-resistant enterococci; MRSA, methicillin-resistant *Staphylococcus aureus*; CoNS coagulase-negative staphylococci; ESBL, extended spectrum beta lactamases; CPE, carbapenem-resistant *Enterobacterales*; MDR, multi-drug-resistant; W1, 21 February–31 July 2020; W2, 1 August 2020–31 January 2021; W3, 1 February–30 September 2021, W4, 1 October 2021 and 30 April 2022.

## Data Availability

The data presented in this study are available on request from the corresponding author. The data are not publicly available due to privacy.

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
