# Peer review of "Bloodstream Infections in Intensive Care Unit during Four Consecutive SARS-CoV-2 Pandemic Waves"

_antibiotics, 2023, doi:10.3390/antibiotics12091448_

Round 1

Reviewer 1 Report

Pozza et al., have an interesting study that is timely and informational. Although the authors' findings are limited by the mass spec-based ID and antimicrobial testing, this manuscript serves as a guide to provide insight and suggests direction for future more in-depth analyses using a multi-omic approach. Nonetheless, I find this manuscript interesting and valuable for publication. Additional minor comments are listed below.

1. Fig1: add y axe label and define periods 1-4 in the legend

2. The authors found a statistical significance of being Italian and there is no discussion about it. What is unique about being Italian with BSI in ICU?

3. Italicize all bacterial species names including P. aeruginosa in line 213

Minor revision is needed. Please see my comments.

Author Response

Pozza et al., have an interesting study that is timely and informational. Although the authors' findings are limited by the mass spec-based ID and antimicrobial testing, this manuscript serves as a guide to provide insight and suggests direction for future more in-depth analyses using a multi-omic approach. Nonetheless, I find this manuscript interesting and valuable for publication. Additional minor comments are listed below.

We thank the Reviewer for his/her kind word.

  1. Fig1: add y axe label and define periods 1-4 in the legend

We thank the Reviewer for the suggestion. We have added the y axes label and provide a legend for the study periods.

  1. The authors found a statistical significance of being Italian and there is no discussion about it. What is unique about being Italian with BSI in ICU?

We thank the reviewer for the observation. The difference in the proportion of subjects admitted in ICU according of being or not Italian was a consequence of the epidemiology of SARS-CoV-2 epidemic in Lombardy. In particular, during W1 the epidemic affected mainly areas outside the metropolitan area of Milan and the proportion of Italians was higher when compared to that of W2 and W3 where most of hospitalized subjects were from the metropolitan area of Milan characterized by a higher proportion of non-Italians. In the end, during W4 when vaccination was widely implemented the proportion of Italian subjects further increased because of the higher median age of subjects with severe consequence of SARS-CoV-2 infection.

  1. Italicize all bacterial species names including P. aeruginosa in line 213

We thank the reviewer for the observation, and we have modified the manuscript accordingly.

Reviewer 2 Report

I congratulate the authors for the paper Bloodstream Infections in Intensive Care Unit During Four 2 Consecutive Sars-Cov-2 Pandemic Waves, an interesting topic. Nice work and the improvements are next:

1. Abstract must be rewritten, reorganized. For instance, backround is not the same with purpose of the study. You must present in the abstract the most important findings of the study.

2. Considering that the waves refer to variable periods such as the number of months, is their comparison correct?

3.etiology of BS is difficult to follow, it was quite difficult to understand  ? numar of enterococcus sp were VRE, or ?number of S aureus were MRSA

4. Discussion must be improved.

5. The susceptibility of strains could be add.

Author Response

I congratulate the authors for the paper Bloodstream Infections in Intensive Care Unit During Four 2 Consecutive Sars-Cov-2 Pandemic Waves, an interesting topic. Nice work and the improvements are next:

We thank the Reviewer for his/her criticisms and suggestion which helped us to significantly improve our manuscript.

Abstract must be rewritten, reorganized. For instance, backround is not the same with purpose of the study. You must present in the abstract the most important findings of the study.

We thank the reviewer for the observation, and we have now provided a revised version of the abstract.

  1. Considering that the waves refer to variable periods such as the number of months, is their comparison correct?

We thank the reviewer for the observation. The reviewer is right, in fact some epidemic periods are longer than other. Nevertheless, by normalizing the incidence of BSIs per patients’ days at risk (days spent in intensive care unit) we were able to compare the difference epidemic periods. The comparison of different epidemic scenario in terms of BSIs incidence is of particular interest if we consider that during the first epidemic period a severe disruption of health care system occurred in Italy and in particular in Lombardy where the study has been carried out. During the following period characterized by differences in terms of pressure poses on hospitals the incidence of BSIs continued to be high in critically ill COVID-19 patients hospitalized in ICU. This was probably due to the characteristics of the disease with a prolonged ICU stay and a severe dysregulation. In addition, the comparison of different period allowed us to observe a progressive reduction of BSIs caused by Enterococcus spp. although still accounting for one out of three of the episodes in the last epidemic periods suggesting underlining pathophysiologic explanation.

3.etiology of BS is difficult to follow, it was quite difficult to understand  ? numar of enterococcus sp were VRE, or ?number of S aureus were MRSA

We thank the reviewer for the observation. We have now clarified the proportion of VRE and MRSA in the result section.

  1. Discussion must be improved.

We thank the Editor and the Reviewer for the suggestion. We have now provided a new improved version of the discussion.

  1. The susceptibility of strains could be add.

We thank the reviewer for the observation. We provided the resistance profile for S.aureus, Enterococcus spp, Enterobacterales and P.aeruginosa ad displayed in Table 2 trying to highlight the main mechanism of resistance (i.e. methicillin resistance and extended spectrum beta lactamases). We have tried to better explain this in the results section.